# Polyurethane Foam Incorporated with Nanosized Copper-Based Metal-Organic Framework: Its Antibacterial Properties and Biocompatibility

**DOI:** 10.3390/ijms222413622

**Published:** 2021-12-19

**Authors:** Do Nam Lee, Kihak Gwon, Yunhee Nam, Su Jung Lee, Ngoc Minh Tran, Hyojong Yoo

**Affiliations:** 1Ingenium College of Liberal Arts (Chemistry), Kwangwoon University, Seoul 01897, Korea; khgwon@kw.ac.kr (K.G.); sue2009@live.co.kr (S.J.L.); 2Department of Physiology and Biomedical Engineering, Mayo Clinic, Rochester, MN 55902, USA; 3Department of Materials Science and Chemical Engineering, Hanyang University, Ansan 15588, Korea; n2021144530@hanyang.ac.kr (Y.N.); tranminhngoc.hueuni@gmail.com (N.M.T.)

**Keywords:** polyurethane foam, nanosized Cu-BTC, castor oil, chitosan, antibacterial, biocompatibility

## Abstract

Polyurethane foams (PUFs) have attracted attention as biomaterials because of their low adhesion to the wound area and suitability as biodegradable or bioactive materials. The composition of the building blocks for PUFs can be controlled with additives, which provide excellent anti-drug resistance and biocompatibility. Herein, nanosized Cu-BTC (copper(II)-benzene-1,3,5-tricarboxylate) was incorporated into a PUF via the crosslinking reaction of castor oil and chitosan with toluene-2,4-diisocyanate, to enhance therapeutic efficiency through the modification of the surface of PUF. The physical and thermal properties of the nanosized Cu-BTC-incorporated PUF (PUF@Cu-BTC), e.g., swelling ratio, phase transition, thermal gravity loss, and cell morphology, were compared with those of the control PUF. The bactericidal activities of PUF@Cu-BTC and control PUF were evaluated against *Pseudomonas aeruginosa*, *Klebsiella pneumoniae*, and methicillin-resistant *Staphylococcus aureus*. PUF@Cu-BTC exhibited selective and significant antibacterial activity toward the tested bacteria and lower cytotoxicity for mouse embryonic fibroblasts compared with the control PUF at a dose of 2 mg mL^−1^. The Cu(II) ions release test showed that PUF@Cu-BTC was stable in phosphate buffered saline (PBS) for 24 h. The selective bactericidal activity and low cytotoxicity of PUF@Cu-BTC ensure it is a candidate for therapeutic applications for the drug delivery, treatment of skin disease, and wound healing.

## 1. Introduction

Polyurethanes (PUs), which can be synthesized by polymerization reactions between isocyanates and polyols, are versatile polymers used in various applications. PUs are omnipresent in daily life owing to their diverse compositions, excellent mechanical properties, and good biocompatibility [1,2,3,4,5]. In general, PUs are prepared from petroleum-based resources; however, with the increasing interest in sustainable and environmental issues, many kinds of sustainable natural materials such as castor oil, crude glycerol, olive stone, lignin, and sorbitol have been used to generate polyols [6,7,8,9,10]. In particular, PUs prepared from plant-based polyols provide additional advantages in biocompatibility for biomedical applications [11,12,13,14]. Therefore, the research area can be expanded to biomedical applications such as catheters, vascular prostheses, breast implants, heart valves, pacemakers, and drug delivery [1,15,16,17,18,19]. Among PU applications, polyurethane foams (PUFs) are one of the most representative products in the PU industry. PUFs can provide adequate moisture and exhibit high gas permeability. Moreover, PUFs have low adhesion to the wound area [20,21,22], and can be functionally designed as either biodegradable or bioactive materials. The composition of the building blocks for PUFs can be controlled with additives, which provide excellent anti-drug resistance and biocompatibility [15,16,17,23].

To enhance the biomedical properties of PUFs, we have attempted to incorporate chitosan and metal–organic frameworks (MOFs) as additives. Chitosan, known as the *N*-deacetylated derivative of chitin, has been used to prepare a variety of polymer architectures such as films, sponges, hydrogels, etc. [24]. Furthermore, it can assume dual roles as a chain extender and an antibiotic toward microorganisms like fungi and bacteria via linkage to an amine group (-NH_2_) [25,26,27,28]. Owing to their high surface area, crystallinity, and easy tunability, crystalline MOFs have been applied in areas such as sensors, gas sorption, separation, and catalysis processes, and have been recently extended to biomedical fields via combination with nanoparticles, antibiotics, and polymeric materials [29,30,31,32,33,34,35,36]. In particular, nanosized MOFs are known to provide a beneficial surface area effect and size dependent properties compared to bulk MOF [37]. Cu-BTC (copper(II)-benzene-1,3,5-tricarboxylate) was used in the current work, since it is utilized in many applications owing to its high porosity and unique structural composition [38,39,40,41,42,43], and exhibited excellent bioactivities toward various microbes; it is frequently applied to various antibacterial polymers [44,45,46,47,48].

Herein, we report the preparation and antibacterial activities of the nanosized Cu-BTC-incorporated PUF (PUF@Cu-BTC), which is composed of nanosized Cu-BTC incorporated with PUFs. PUF@Cu-BTC was prepared by crosslinking castor oil and chitosan with toluene-2,4-diisocyanate (TDI) in the presence of nanosized Cu-BTC and water. This is an example of the synergistic combination of sustainable PUFs based on castor oil, antibacterial chitosan, and nanosized Cu-BTC, which can be used for biomedical and therapeutic applications. The structures of the prepared materials were fully characterized, and their antibacterial activities against *Pseudomonas aeruginosa* (*P. aeruginosa*), *Klebsiella pneumonia* (*K. pneumonia*), and methicillin-resistant *Staphylococcus aureus* (MRSA) were tested. The biocompatibility of the materials in vitro was also examined in mouse embryonic fibroblasts.

## 2. Results and Discussion

### 2.1. Preparation of PUF and PUF@Cu-BTC

Cu-BTC, formulated as Cu_3_(BTC)_2_ (copper(II)-benzene-1,3,5-tricarboxylate), was synthesized following a previously reported method, albeit, incorporating a slight modification [49]. Typically, Cu(NO_3_)_2_∙3H_2_O reacted with H_3_BTC in mixed solvent of ethanol, deionized water, and DMF at 80 °C for 24 h. Nanosized Cu-BTC was then prepared by ball-milling the as-prepared Cu-BTC with a weight ratio of 50:1 (balls-to-Cu-BTC) for better dispersion derived from the small size of the polymer solution.

PUF@Cu-BTC was prepared as a film via foaming reaction of castor oil (polyol), chitosan (chain extender), nanosized Cu-BTC (antibacterial additive), Dabco 33 (catalyst), and water (foaming agent) with TDI in 10% excess molar ratio. The foaming processes conventionally proceed via multi-step reactions including urethane, urea, allophanate, and biuret formations as shown as Figure 1 [50]. In detail, TDI reacts with castor oil and water to form typical urethane and urea linkage evolving CO_2_ foaming gas, respectively. Amines derived from water and chitosan further react with isocyanate to extend polymer chains through urea bond, and then, urethane and urea are converted to allophanate and biuret linkage by secondary reactions with isocyanate, respectively. Foam density of PUF@Cu-BTC was measured as higher, 0.69 g mL^−1^ than that of hard polyurethane foams [51]. The PUF was also synthesized as a control experiment via the same protocol as that for PUF@Cu-BTC, but excluding the nanosized Cu-BTC.

### 2.2. Characterizations of Nanosized Cu-BTC, PUF, and PUF@Cu-BTC

The crystalline natures of Cu-BTC, nanosized Cu-BTC, PUF, and PUF@Cu-BTC were determined by powder X-ray diffraction (PXRD) in range of 0 to 90°. The PXRD pattern of Cu-BTC was in line with the simulated pattern from the X-ray crystallographic data (Appendix A). After ball-milling, the crystallinity of the nanosized Cu-BTC decreased compared with Cu-BTC. PUF and PUF@Cu-BTC were observed as semi-crystalline polymers with a broad peak at 20° (Figure 1).

From TEM images, nanosized Cu-BTC was fabricated as irregular lumps comprising an agglomerated powder in the range of 100 to 200 nm (Figure 2).

In Fourier-transform infrared spectroscopy (FT-IR), both PUF and PUF@Cu-BTC exhibit broad N-H absorption band near 3344 cm^−1^ and the absorption band of excess isocyanate (-N=C=O) appeared at 2274 cm^−1^ [52]. The sharp absorption at 1730 cm^−1^ corresponds to typical symmetric stretching bands ν(C=O). The weak adsorption of C-N stretching was observed at 1524 cm^−1^, while the stretching mode ν(C=O) for the carbonyl of Cu-BTC was not observed at 1640 cm^−1^ due to its low concentration and overlapping with peaks derived from urethane linkage (-NH-COO-) (Figure 3).

Differential scanning calorimetry (DSC) and thermogravimetric analysis (TGA) were performed to investigate the thermal stability of nanosized Cu-BTC on PUF (Figure 4).

The glass transition temperature (T_g_) did not appear in the DSC curves of both PUF and PUF@Cu-BTC because of their flexibility. However, there is an endo peak at approximately 320 °C and an exo peak at approximately 425 °C in the DSC curves of the two samples corresponding to the hard segment crystalline melting points and the degradation points, respectively (Figure 4a).

The TGA curves illustrate the two-step degradation of PUF and PUF@Cu-BTC. In the first step, the urethane bond (-NH-CO-O-) was smoothly broken from 300 to 500 °C (approximately 60%). In the second step, the urethane linkages (-HN-CO-NH) and polyol bonds were cleaved in the temperature range of 500 to 580 °C because of the higher bond strength (approximately 37%) [53]. Additionally, the degradation of Cu-BTC was weakly observed in the PUF-Cu-BTC plot from 350 to 500 °C. Moreover, PUF@Cu-BTC decomposed slightly less than PUF (approximately 1%) because of the thermal stability of the CuO derived from Cu-BTC (Figure 4b).

Figure 5 presents representative SEM images and the corresponding elemental distribution spectra (EDS) elemental mapping data for PUF and PUF@Cu-BTC.

The C, O, and N elements were distributed evenly throughout the network of the two samples. In addition, the presence of nanosized Cu-BTC in PUF@Cu-BTC was further confirmed by EDS mapping of Cu. The micrograph illustrates that PUF@Cu-BTC comprises more cells in smaller sizes than those in PUF, and the average cell size of the foam was decreased from 350 to 50 μm after Cu-BTC was incorporated into PUF (Appendix A). This phenomenon implies that CO_2_ gas is necessary for cell growth, and possibly absorbs into the porous Cu-BTC or bonds on its active metal sites, resulting in a decreased cell size and slightly increased density than PUF (0.69 vs. 0.64 g mL^−1^) [31,43].

### 2.3. Swelling Properties of PUF@Cu-BTC

PUF and PUF@Cu-BTC’s swelling ratios were observed as 1.11 ± 0.06 and 1.22 ± 0.04, respectively (Table 1). Meanwhile, PUF@Cu-BTC exhibited a slightly larger swelling ratio compared to that of PUF, which can be explained by the porosity of Cu-BTC which allows for favorable water storage owing to the smaller cells. From these results, PUF@Cu-BTC was considered to maintain polymeric structures similar to those of the control PUF.

### 2.4. Bactericidal Test

The bactericidal properties of PUF and PUF@Cu-MOF were tested for potential therapeutic applications; the antibacterial properties of the antibacterial chitosan-based PUF were tested against *P. aeruginosa*, *K. pneumoniae*, and MRSA (Figure 6 and Table 2).

Bactericidal percentage increased in the following order: MRSA (30.8%), *P. aeruginosa* (66.3%), and *K. pneumoniae* (99.3%). PUF exhibited considerable antibacterial properties against *K. pneumoniae,* and this may be attributed to the inclusion of chitosan [25,26]. When Cu-BTC was incorporated into PUF (PUF@Cu-BTC)***,*** its bactericidal property increased significantly toward MRSA (30.87 to 77.6%) and *P. aeruginosa* (66.3 to 97.8%), but the bactericidal percent toward *K. pneumoniaee* increased to 99.9% at 2 mg/mL, where elimination of all bacteria is preferred (99.9%), instead of natural selection leading to resistance (99.3%) [53,54,55,56,57]. Both PUF and PUF@Cu-BTC had the highest bioactivity against *K. pneumoniae*.

In conclusion, these results indicate that PUF@Cu-BTC exhibits a synergistic bactericidal effect derived from the combination of chitosan and Cu-BTC on MRSA and *P. aeruginosa.*

### 2.5. Ion Release Test

The degradation tests of PUF@Cu-BTC were performed for 6, 12, and 24 h in PBS at 25 °C, respectively. The amount of copper (II) ions released from PUF@Cu-BTC was observed using inductively coupled plasma mass spectrometry (ICP-MS). Although the concentration of metal ions released from 1 mg mL^−1^ of PUF @Cu-BTC increased to 6 h and then suddenly reduced as 9 ppb until 24 h (Figure 7). Conversely, the concentration of Cu(II) ions released from Cu-BTC was observed as 5366 ppb at 24 h, which was over 596 times higher than that of PUF@Cu-BTC (9 ppb). Cu-BTC presents synergistic bactericidal property against all bacterial strains, while it stays stably at PUF network releasing very lower concentration of Cu (II) ion than Cu-BTC.

### 2.6. Cytotoxicity of PUF@Cu-BTC

Based on prior study, we investigated the cell biocompatibility of PUF@C-BTC [58,59]. To verify the toxicity of the control PUF and PUF@Cu-BTC, each sample was prepared using the same method. Furthermore, as an additional positive control (blank), an MEF monolayer with no contact was prepared. An MEF monolayer contact with a 10% EtOH solution was employed as a negative control. Both live/dead staining and a colorimetric MTS test were used to monitor MEF viability. The cell viabilities for the blank, PUF, and PUF@Cu-BTC samples exceeded 95% at 1 day after culture, as demonstrated in the fluorescence microscopy images (Figure 8a), but the majority of cells died after exposure to EtOH.

After seeding, the cells gradually attached, spread, and flattened on the surface before creating a confluent layer that lasted for 3 days; however, this was not observed in the EtOH group (data were not shown). These results indicate that PUF and PUF@Cu-BTC were not harmful to the cells. An MTS assay was used for further quantification (Figure 8b). PUF and PUF@Cu-BTC extracted from cell culture media solutions were serially diluted. For example, 100% denotes the original extraction medium, while 25% denotes a fourfold dilution of the original extraction media. Following the creation of the MEF monolayer, the culture media was replaced to a media containing the extraction solution at the required concentration and cultivated. The acquired MTS results showed that PUF@Cu-BTC had good cytocompatibility, with MEF viability over 95% in all the cases; however, most MEFs died when exposed to EtOH, validating the low cytotoxicity of the PUF and PUF@Cu-BTC extracts.

## 3. Materials and Methods

### 3.1. Preparation of Nanosized Copper(II)-Benzene-1,3,5-Tricarboxylate (Cu-BTC)

Cu-BTC was prepared by a previously reported solvothermal method utilizing copper(II) ions and 1,3,5-benzenetricarboxylate ligands [49]. In a 20 mL vial, copper(II) nitrate trihydrate (Cu(NO_3_)_2_∙3H_2_O, 99%, Acros, Seoul, Korea) (0.725 g, 3.0 mmol) was dissolved in deionized water (10 mL). Meanwhile, 1,3,5-benzenetricarboxylic acid (H_3_BTC, C_9_H_6_O_6_, 98%, Acros, Seoul, Korea) (0.210 g, 1.0 mmol) was dissolved in ethanol (10 mL). The Cu(NO_3_)_2_ solution was quickly added to the H_3_BTC solution at 25 °C and stirred and then, DMF (0.7 mL) was added into this solution. The reaction mixture was rose to 80 °C and placed for 24 h at this temperature. After cooling naturally to 25 °C, the resultant product was collected through centrifuging and dried under vacuum for 24 h after several times washing with deionized water and ethanol.

Nanosized Cu-BTC was prepared using a ball-milling method. The as-prepared Cu-BTC was ball-milled at 350 rpm for 6 h at a weight ratio of 50:1 (balls-to-Cu-BTC) using a Fritsch™ Planetary Mill Pulverisette 5 (Fritsch GmbH, Idar-obertein, Germany).

### 3.2. Preparation of PUF and Nanosized Cu-BTC Incorporated Polyurethane Foam (PUF@Cu-BTC)

PUF@Cu-BTC was prepared by crosslinking a mixture of castor oil (10 g, H-35, Itoh Oil Chemicals Co., LTD, Yokkaichi, Japan), chitosan (160 mg, Aldrich, Darmstadt, Germany), nanosized Cu-BTC (15 mg), 1,4-diazabicyclo[2.2.2]octane (Dabco 33, 30 mg, Aldrich, Darmstadt, Germany), and water (100 mg) with toluene-2,4-diisocyanate (TDI, Aldrich, Darmstadt, Germany) in 10% excess molar ratio. In detail, a mixture of castor oil (10 g), chitosan (160 mg), Dabco 33 (30 mg), and water (100 mg) was first mixed with nanosized Cu-BTC (15 mg). Subsequently, TDI was added to the above mixture, and the obtained suspension was pipetted into a silicone rubber mold for vulcanizing for 24 h at 25 °C.

For the control experiment, PUF was prepared using the same protocol, but, excluding nanosized Cu-BTC.

### 3.3. Instrumentation

PXRD patterns were obtained using a Rigaku MiniFlex diffractometer (Neu-Isenburg, Germany; 30 kV, 15 mA, scan speed: 2° min^−^^1^, step size: 0.02°). FTIR spectra were measured on a Nicolet iS10 FTIR spectrometer with KBr pellets (Thermo Fisher Scientific, Waltham, Massachusetts, USA). DSC and TGA was performed using DSC (DSC 214 polyma, NETZSCH, Burlington, MA, USA), and TGA (TG 209 F3 Tarsus^®^, NETZSCH, Burlington, MA, USA), respectively. The surface morphology and elemental composition of PUFs were characterized using SEM-EDS (FE-SEM, JEOL JSM-5800F, Peabody, MA, USA) and TEM (JEOL JEM-2100F, Peabody, MA, USA). Degradation of PU@Cu-BTC was tested by inductively coupled plasma mass spectrometry (NexION 350D, Perkin-Elmer SCIEX, Waltham, MA, USA). The fluorescence intensity was read with a microplate reader (Synergy H1, BioTek, Winooski, VT, USA) and the stained cells were imaged by a fluorescent microscope (IX83, Olympus, Center Valley, PA, USA).

### 3.4. Swelling Ratio of PUF@Cu-BTC

PUF and PUF@Cu-BTC were swollen for 48 h in 0.01 M PSB for the swelling ratio tests. The residual moisture on the surface of fully swollen PUF and PUF@Cu-BTC samples was drained using filter paper after they were removed from the PBS solution. The swollen samples were individually weighed (Ws) and freeze-dried and weighed (Wd), and their swelling ratios were calculated using the following equation (Equation (1)):Swelling ratio = Ws/Wd (*n* = 4) (1)

### 3.5. Degradation and Metal Ion Release Test

Three test solutions consisting of PUF@Cu-BTC were prepared for the ion leakage test at a concentration of 1 mg mL^−1^ in PBS and stirred for 6, 12, and 24 h at 25 °C. Nanosized Cu-BTC (1 mg mL^−1^) was evaluated using the same method. The quantity of Cu(II) ions released in the samples was measured by ICP-MS with the supernatant separated from each test tube after centrifugation. The degree of degradation is expressed as ppb (μg Kg^−1^), and the concentration of Cu(II) ions released into the medium at each release test time.

### 3.6. Antibacterial Test

According to a previously published test method, the antibacterial properties of PUF and PUF@Cu-BTC were assessed against three bacteria strains: *P. aeruginosa*, *K. pneumoniae*, and MRSA [60]. In brief, the antibacterial properties of PUF@Cu-BTC were tested against three strains of bacteria, three specimens of 50 × 50 mm^2^ (within 10 mm thickness) and a Stomacher film of the same size was prepared as a control. As a positive control, PUF (without Cu-BTC) was made and tested. The test specimen’s surface was wiped with ethanol 2−3 times using gauze before being completely dried. Platelets containing precultured test bacteria were routinely inoculated with 10^5^~10^6^ colony-forming units (cfu)·mL^−1^. The test side of each test component was placed in a Petri dish. Using a pipette, 0.2 mL of the test solution was inoculated onto each test piece. To disseminate the test bacteria over the film, the film on the fallen test bacterium was covered and lightly squeezed. The test piece inoculated with the test strain and the control Petri dish were incubated at 37 °C for 24 h. To wash away the test bacteria, 10 mL of SCDLP medium was added. The viable cells were counted using this washed solution. After incubation, the test bacteria were washed off. The washing solution (1 mL) was added to 9 mL of physiological saline and mixed completely. The washing solution was diluted by step by step as described in this procedure, and the 100 μL of the diluted solution was plated onto three nutrient agar plates and incubated at 37 °C for 24 h. All of the experiments were carried out in three times.

### 3.7. Cytotoxicity Assays

Cytotoxicity of PUF and PUF@Cu-BTC was assessed as described in prior papers [58]. MEFs were maintained at 37 °C in a humidified incubator with 5% CO_2_ in T75 flasks containing DMEM (supplemented with 10% (*v*/*v*) FBS, 200 IU·mL^−^^1^ penicillin, and 200 μg·mL^−^^1^ streptomycin). On the collagen type I-coated glass slide (1.25 cm × 1.25 cm), MEFs were seeded at a density of 5 × 10^4^ cells·cm^−1^ and cultured for 3 h. Non-adherent cells were rinsed in PBS, transferred to a well plate, and cultured. Each PUF and PUF@Cu-BTC (1.5 cm diameter) was carefully placed on the cell monolayer at the next day. The incubation period was 24 h. The live/dead assay (ThermoFisher Scientific, Waltham, MA, USA) was used to examine the cells, with the live cells fluorescing green and the dead cells fluorescing red. Inverted fluorescence microscopy (IX83, Olympus, Center Valley, PA, USA) was used to examine the labeled cells after they had been washed in PBS. The ratio of living cells to the total number of cells was used to determine cell viability. In addition, the viability of cells was quantified using an MTS test, which assesses the metabolic rates. PUF and PUF@Cu-BTC were incubated in a cell culture medium for 24 h to obtain the extract solutions, and the cells were seeded on a 24-well plate at a density of 5 × 10^4^ cells per well, as stated in earlier reports. The culture media in each well was changed with DMEM containing the extract solution (200 μL) after 24 h of incubation at 37 °C [59]. The DMEM-containing extract solution was carefully removed after 24 h of incubation, and each well was subsequently filled with an MTS cell proliferation assay kit solution (20 μL) and a fresh media (200 µL). The absorbance at a wavelength of 490 nm was measured using a microplate reader after an additional 4 h of incubation (Synergy H1, BioTek, Winooski, VT, USA). The number of proliferating cells was determined using the following equation and a standard curve of cells (Equation (2)):Cell viability (%) = (OD_sample −_ OD_blank_/OD_control −_ OD_blank_) × 100 (2)
where the absorbance of the wells containing extract solution was OD sample, the absorbance of the wells containing only culture media was OD control, and the absorbance of the wells without cell was OD blank.

## 4. Conclusions

In this study, nanosized Cu-BTC was successfully incorporated into the PUF surface via a crosslinking process at 25 °C to afford PUF@Cu-BTC. The physical and thermal properties seemed to be similar to those of the as-prepared PUF. The release test of Cu(II) ions indicated that PUF@Cu-BTC was stable in a PBS solution, and PUF@Cu-BTC exhibited significant antibacterial activities and selectivity against three strains of bacteria, *P. aeruginosa, K. pneumoniae*, and MRSA, and low cytotoxicity in MEFs at 2 mg mL^−1^. Furthermore, PUF@Cu-BTC exhibited an excellent synergistic effect for bactericidal properties when combined with chitosan and Cu-BTC, compared with PUF. The low cytotoxicity and high bactericidal activity of PUF@Cu-BTC indicate its suitability as a candidate for therapeutic applications for the drug delivery, treatment of skin disease, and wound healing.

## Data Availability

Not applicable.

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
