# Peer review of "Polyurethane Foam Incorporated with Nanosized Copper-Based Metal-Organic Framework: Its Antibacterial Properties and Biocompatibility"

_ijms, 2021, doi:10.3390/ijms222413622_

Round 1

Reviewer 1 Report

With joy I read the manuscript entitled "Polyurethane Foam Incorporated with Nano-Sized Copper-Based Metal-Organic Framework: Its Antibacterial Properties and Biocompatibility" by Lee and colleagues. The research was carried out rigorously. Also, a SI is available containing the SEM and XRD data. To me, the manuscript can be published as is.

Author Response

Reviewer #1

With joy I read the manuscript entitled "Polyurethane Foam Incorporated with Nano-Sized Copper-Based Metal-Organic Framework: Its Antibacterial Properties and Biocompatibility" by Lee and colleagues. The research was carried out rigorously. Also, a SI is available containing the SEM and XRD data. To me, the manuscript can be published as is.

Response: We sincerely appreciate the reviewer #1 for reviewing our paper.

Reviewer 2 Report

Ms. Ref. No.: ijms-1485812
Title: Polyurethane Foam Incorporated with Nano-Sized Copper-2 Based Metal-Organic Framework: Its Antibacterial Properties 3 and Biocompatibility

Reviewer comments

It is a well-planned and executed piece of research. Results are clearly presented, and findings are sound and well-reasoned. Polyurethanes are essential materials in the medical field, and any improvements to their properties are always welcome.

Specific comments

Line 174: “When Cu-BTC was incorporated into PUF (PUF@Cu-BTC), its bactericidal property increased significantly toward MRSA (30.8%–77.6%) and P. aeruginosa (66.3%–97.8%),…” I’d suggest using an arrow to indicate the increase, e.g., 30.8%→77.6%. The marginal increase from 99.3% to 99.9% might not be as slight as the authors think, in the context of “drug-resistant” strains of bacteria, where elimination of all bacteria is preferred (99.9%), instead of natural selection leading to resistance (99.3%). Perhaps there is some information in that context is available in the literature?

Author Response

Reviewer #2

“When Cu-BTC was incorporated into PUF (PUF@Cu-BTC), its bactericidal property increased significantly toward MRSA (30.8%–77.6%) and P. aeruginosa (66.3%–97.8%),…” I’d suggest using an arrow to indicate the increase, e.g., 30.8%→77.6%. The marginal increase from 99.3% to 99.9% might not be as slight as the authors think, in the context of “drug-resistant” strains of bacteria, where elimination of all bacteria is preferred (99.9%), instead of natural selection leading to resistance (99.3%). Perhaps there is some information in that context is available in the literature?

Response: We thank reviewer for your important comment. We changed dash to arrow and modified the context based on your advice as following with additional reference: When Cu-BTC was incorporated into PUF (PUF@Cu-BTC), its bactericidal property increased significantly toward MRSA (30.8%→77.6%) and P. aeruginosa (66.3%→97.8%), but the bactericidal percent toward K. pneumoniaee increased to 99.9% at 2 mg/mL, where elimination of all bacteria is preferred (99.9%), instead of natural selection leading to resistance (99.3%) [54].  

Reviewer 3 Report

The paper Polyurethane Foam Incorporated with Nano-Sized Copper-Based Metal-Organic Framework: Its Antibacterial Properties and Biocompatibility, prepared by Do Nam Lee et al., presents novel and interesting results that deserve to be published after some minor improvements:

  1. please highlight the nanometric size by adding TEM images.
  2.  please add absorbance values for FT-IR
  3. Figure 6 - please insert statistical bars.

Author Response

Reviewer #3

The paper Polyurethane Foam Incorporated with Nano-Sized Copper-Based Metal-Organic Framework: Its Antibacterial Properties and Biocompatibility, prepared by Do Nam Lee et al., presents novel and interesting results that deserve to be published after some minor improvements:

  1. please highlight the nanometric size by adding TEM images.

Response: We thank reviewer for raising this point. We test TEM and changed SEM images with TEM images as following: From TEM images, nano-sized Cu-BTC was fabricated as irregular lumps comprising an agglomerated powder in the range of 100 nm to 200 nm (Figure 2).

Figure 2. Transmission electron microscopy (TEM) images of nano-sized Cu-BTC. Scale bar: 100 nm, 200 nm, and 0.5 μm, respectively.

  1. please add absorbance values for FT-IR.

Response: We thank reviewer for raising this point. We added the values of absorbance and revised the sentences like this: In Fourier-transform infrared spectroscopy (FT-IR), both PUF and PUF@Cu-BTC exhibit broad N-H absorption band near 3344 cm-1 and the absorption band of excess isocyanate (-N=C=O) appeared at 2274 cm-1 [52]. The sharp absorption at 1730 cm-1 corresponds to typical symmetric stretching bands n(C=O). The weak adsorption of C-N stretching was observed at 1524 cm-1, while the stretching mode n(C=O) for the carbonyl of Cu-BTC was not observed at 1640 cm-1 due to its low concentration and overlapping with peaks derived from urethane linkage (-NH-COO-) (Figure 3).

Figure 3. Fourier-transform infrared spectroscopy spectra of nano-sized Cu-BTC (black), PUF (blue), and PUF@Cu-BTC (red).

  1. Figure 6 - please insert statistical bars.

Response: We thank reviewer for this comment. We modified it as following as

Figure 6. Antibacterial efficiency of PUF and PUF@Cu-BTC towards Pseudomonas aeruginosa, Klebsiella pneumoniae, and methicillin resistant Staphylococcus aureus (Mean ± standard deviation with n = 3; NS: not significant, ** p < 0.01, *** p < 0.001.)
